

# Identification of the biological function of miR-9 in spinal cord ischemia-reperfusion injury in rats

Fengshou Chen, Jie Han, Xiaoqian Li, Zaili Zhang and Dan Wang

Department of Anesthesiology, the First Hospital of China Medical University, Shenyang, Liaoning, China

## ABSTRACT

Spinal cord ischemia–reperfusion injury (SCII) is still a serious problem, and the mechanism is not fully elaborated. In the rat SCII model, qRT-PCR was applied to explore the altered expression of miR-9 (miR-9a-5p) after SCII. The biological function of miR-9 and its potential target genes based on bioinformatics analysis and experiment validation in SCII were explored next. Before the surgical procedure of SCII, miR-9 mimic and inhibitor were intrathecally infused. miR-9 mimic improved neurological function. In addition, miR-9 mimic reduced blood-spinal cord barrier (BSCB) disruption, inhibited apoptosis and decreased the expression of IL-6 and IL-1β after SCII. Gene Ontology (GO) analysis demonstrated that the potential target genes of miR-9 were notably enriched in several biological processes, such as "central nervous system development", "regulation of growth" and "response to cytokine". The Kyoto Encyclopedia of Genes and Genomes (KEGG) analysis revealed that the potential target genes of miR-9 were significantly enriched in several signaling pathways, including "Notch signaling pathway", "MAPK signaling pathway", "Focal adhesion" and "Prolactin signaling pathway". We further found that the protein expression of MAP2K3 and Notch2 were upregulated after SCII while miR-9 mimic reduced the increase of MAP2K3 and Notch2 protein. miR-9 mimic or MAP2K3 inhibitor reduced the release of IL-6 and IL-1β. miR-9 mimic or si-Notch2 reduced the increase of cleaved-caspase3. Moreover, MAP2K3 inhibitor and si-Notch2 reversed the effects of miR-9 inhibitor. In conclusion, overexpression of miR-9 improves neurological outcomes after SCII and might inhibit BSCB disruption, neuroinflammation, and apoptosis through MAP2K3-, or Notch2-mediated signaling pathway in SCII.

Corresponding author
Dan Wang,
wonder12251@hotmail.com

## INTRODUCTION

Spinal cord ischemia reperfusion injury (SCII) is a complication occurring under thoracoabdominal aortic or spinal cord surgery, which brings the risk of paralysis and paraplegia (*Li et al., 2015a*; *Xu & Li, 2020*). Although various measures were developed and applied to reduce the risks and improve the outcomes of SCII, satisfactory therapeutic

effects still not be achieved due to multifactorial pathogenic factors (*Ege et al., 2004*; *Fang et al., 2015*; *Li et al., 2014a*; *Li et al., 2014b*; *Zhou et al., 2013b*).

MicroRNAs (miRNAs, miRs) are 21- to 23-nucleotide noncoding RNA that are capable of specific binding mRNA and regulate post-transcriptional expression (*Chen et al., 2020a*; *He et al., 2015*; *Li et al., 2016b*). It has been discovered that miRNAs might be related to central nervous system (CNS) injury including SCII (*Bao et al., 2018*; *Chi et al., 2014*; *Huang et al., 2015*; *Li et al., 2015b*; *Yao, Wang & Zhang, 2018*). miR-27a mimics reduced blood-spinal cord barrier (BSCB) damage induced by neuroinflammation following SCII via inhibiting the NF-κB/IL-1β pathway and negatively regulating the TICAM-2 of the TLR4 signaling pathway (*Li et al., 2015b*). In rat models of SCII, miR-125b mimic was found to protect against SCII via reducing aberrant p53 network activation-induced apoptosis and neuroinflammation through the downregulation of TP53INP1 (*Li et al., 2018c*). Several studies have demonstrated altered miRNA expression profiles in spinal cord tissues of SCII models, which implicated the important roles of miRNAs in the pathophysiological mechanism of SCII (*Hu, Lv & Yin, 2013*; *Li et al., 2016a*; *Liu et al., 2020*).

miR-9 (miR-9a-5p) is a highly conserved mature miRNA across species and serves as a tumour regulator in several cancer types, including colorectal cancer, gastric cancer and breast cancer (*Gao et al., 2017*; *Wang et al., 2015*; *Yang et al., 2019*; *Zhu et al., 2015*). Evidence also showed that miR-9 is highly expressed in the brain, that is involved in ischemic stroke, transient cerebral ischemia, amyotrophic lateral sclerosis (ALS) (*Altintas et al., 2016*; *Cao et al., 2020*; *Hawley, Campos-Melo & Strong, 2019*). Knockdown of TUG1 promoted cell survival and decreased cell apoptosis through increasing the expression of miR-9 and inhibiting the expression of Bcl2 following brain ischemia(*Chen et al., 2017*). A recent study has shown that ferulic acid treatment protected against neuronal death in the rat hippocampus following hypoxic-ischemic damage through the inhibition of miR-9 induction (*Yao et al., 2020*). However, whether miR-9 exerts an effect on SCII is unknown.

The aim of the present study was to explore the biological function of miR-9 in SCII and clarify the mechanism via combining bioinformatics analysis and experiment validation.

## MATERIALS AND METHODS

### Experimental animals

Sprague–Dawley rats (male, 200–250g) were obtained in the present study. All rats were maintained for at least 1 week before the surgical procedures, with freely available rodent chow and water at 22–24 °C and 50–60% relative humidity, under a 12h/12h light-dark cycle. The present study had approval from the Ethics Committee of China Medical University, Shenyang (CMU 2020391), and were carried out in conformity with the National Institutes of Health Guide for the Use and Care of Laboratory Animals (NIH Publications No.80-23, revised 1996).

## Rat model

To create SCII rat models, a cross-clamped aortic arch was used as previously reported (*Chen et al., 2020a*; *Chen et al., 2020b*). Pentobarbital sodium (50 mg/kg) was intraperitoneal injected for anesthetizing rats. SCII was induced by the aorta occlusion for 14 min. The same surgical procedures without occlusion were performed on sham-operated rats.

## Euthanization

Rats were euthanized by sevoflurane overdose and ensured that it was effective by pinching the tail with tweezers. Any movement in the rat showed that pain could still be felt so enough time was allowed for the anesthesia to fully work before sacrificing the mice. Rats that survived the study or were excluded were bred for other experiments.

## Quantification of miRNA expression

Total RNA was extracted from segments L4–L6 of the spinal cord with using Trizol reagent (Takara, Otsu, Japan). RNA was reverse-transcribed into cDNA using the Prime Script® miRNA cDNA Synthesis Kit (Takara, Tokyo, Japan) (*Li et al., 2016b*; *Wang et al., 2015*). The levels of miR-9 were detected using SYBR Premix qRT-PCR on a PCR System (Corbett Research, Australia) (*Chen et al., 2020a*). The primers used for miR-9 in the present study were as follows: forward: 5′-CGCGCTCTTTGGTTATCTAGCTGTATG-3′ (*Yao, Wang & Zhang, 2018*). Relative miR-9 expression was normalized to U6 expression levels using the $2^{-\Delta\Delta Ct}$ method.

## Intrathecal administration

Thoracic laminectomy was performed for intrathecal pretreatment (*Li et al., 2015b*). Pretreatment with a synthesized miR-9 mimic (5′-UCUUUGGUUAUCUAGCUGU AUGA-3′), inhibitor (5′-TCATACAGCTAGATAACCAAAGA-3′) and negative control (NC) was previously described (*Li et al., 2015b*). Rats were intrathecally injected with 10 µl of the oligonucleotides (500 pmol/10 µl) and EntransterTM-in vivo transfection regent (Engreen, Beijing, China) (*Wang et al., 2020*). Intrathecal injection was applied in vivo prior to ischemia induction once a day for three consecutive days according to the results of our preliminary experiment(*Li et al., 2016b*; *Li et al., 2015b*). In order to inhibit the expression of MAP2K3, SB203580 (5 µl, dissolved in 0.1 nmol/µl solution using 1% DMSO) once daily for 2 consecutive days (*Yao, Wang & Zhang, 2018*; *Zhou et al., 2016*) at the same time. Notch2 siRNA or NC RNA were provided by Jima Inc. (Shanghai, China). For suppressing the expression of Notch2, two days before ischemia, intrathecal infusion of 5 µg of siRNA(5′- CCTCCCATCGTGACTTTCCAGCTTA-3′)or control RNA at a concentration of 1 µg/µl once a day was carried out, as directed by the manufacturer (*Chen et al., 2020b*; *Meng et al., 2015*; *Wang et al., 2020*).

## Experimental protocol

### Protocol I

To measure expression of miR-9 at various time points (24, 48 and 72 h) after reperfusion, rats were euthanized at each time point after SCII ($n = 3$). Rats were assigned to four

groups ($n = 8$): (1) Sham group; (2) SCII group; (3) SCII+NC group; (4) miR-9 mimic group. miR-9 NC or miR-9 mimics was injected intrathecally prior to SCII induction once a day for three consecutive days in the SCII+NC group or miR-9 mimic group, respectively (*Li et al., 2015b*; *Wang et al., 2020*).

### Protocol II

Rats were assigned to eight groups ($n = 8$): (1) Sham group; (2) SCII group; (3) SCII+NC group; (4) miR-9 mimic group; (5) miR-9 inhibitor group; (6) SB203580 + miR-9 inhibitor group; (7) SB203580 group; (8) DMSO group. In the SB203580 + miR-9 inhibitor group, except for miR-9 inhibitor pretreatment, SB203580 (5 μl, dissolved in 0.1 nmol/μl solution using 1% DMSO) once daily was injected intrathecally once a day for two consecutive days before SCII (*Yao, Wang & Zhang, 2018*; *Zhou et al., 2016*). SB203580 or DMSO was injected intrathecally prior to SCII induction once a day for two consecutive days in the SB203580 group or DMSO group, respectively.

### Protocol III

Rats were assigned to eight groups ($n = 8$): (1)–(5) The first five groups are the same as those in Protocol II; (6) si-Notch2 + miR-9 inhibitor group; (7) si-Notch2 group; (8) siRNA NC group. In the si-Notch2 + miR-9 inhibitor group, except for miR-9 inhibitor pretreatment, si-Notch2 5 μg at a concentration of 1 μg/μl was injected intrathecally once a day for two consecutive days before SCII except for miR-9 inhibitor pretreatment (*Chen et al., 2020b*; *Meng et al., 2015*; *Wang et al., 2020*). si-Notch2 or siRNA NC was injected intrathecally prior to SCII induction once a day for two consecutive days in the si-Notch2 group or siRNA NC group, respectively.

## Neurological evaluation

Two observers evaluated the movement function of the rat lower limb based on the BBB scoring after SCII as described previously (*Bao et al., 2018*; *Basso, Beattie & Bresnahan, 1995*).

## Bioinformatics analysis for potential target genes of miR-9

At first, targets of miR-9 were predicted by means of databases miRDB and Targetscan. Target genes were identified for miR-9 in both databases. Then the raw data of GEO Series (GSE)138966 [species: *Rattus norvegicus*; Platforms: GPL22396Illumina HiSeq 4000 (*Rattus norvegicus*)] was obtained from Gene Expression Omnibus database (http://www.ncbi.nlm.nih.gov/geo/). 3 sham-operated samples and 3 SCII samples at 48 h post-SCII were included in GSE138966 (*Ding et al., 2020*). The differentially expressed genes (DEGs) between sham-operated samples and SCII samples were obtained. DEGs are those genes with an $|log2FC| \geq 1$ and $p < 0.05$. For the purpose of analyzing the diversification of DEGs expression, the heatmaps of DEGs were drawn by heatmap function. We chose the intersection of the up-regulated DEGs and the above predicted genes as the potential miR-9 target genes and conducted bioinformatics analysis.

DAVID 6.8 (https://david.ncifcrf.gov/) is a database with annotation, visualization, and integrated discovery functions (*Huang, Sherman & Lempicki, 2009*). The annotation table
is its main analysis tool, which contains functional annotation charts, functional annotation clustering, and functional annotation table subtools (*Wang et al., 2021*). Gene Ontology (GO) and Kyoto Encyclopedia of Genes and Genomes (KEGG) annotation of DEGs was carried out through the annotation tool. The biological functional coherence and biological attributes of the putative target genes were determined based on GO analysis. KEGG is a collection of databases dealing with genomes, diseases, biological pathways, drugs and chemical materials (*Altintas et al., 2016*). The biological processes and pathways of potential miR-9 target genes were analyzed using the DAVID online database. $p < 0.05$ was selected as the cut-off criterion with statistic difference. According to the relations of genes and statistically significant pathways as well as the relations among genes, pathways and miR-9,we also built a miRNA–pathway–gene network

### Evans blue (EB) extravasation

EB cannot normally pass through BSCB and thus its presence in spinal cord tissue indicates BSCB disruption (*Goldim, Della Giustina & Petronilho, 2019*). At the time point of 48 h after SCII, the intravenous injection of EB (45 mg/kg) was performed. The rats were euthanized after 1 h circulation. Segments L4–L6 were homogenized with trichloroacetic acid, and the tissues were centrifuged (*Li et al., 2015b*). The absorption of the supernatant was measured at 632 nm with a microplate reader (BioTek, Winooski, USA) (*Fang et al., 2015*). EB staining was visualized using a fluorescent microscopy (Leica, German) with a green filter (*Chen et al., 2020a*).

### Water content of spinal cord

Water content (%) of segments L4–L6 was obtained using the following formula: (wet weight − dry weight)/wet weight × 100% (*Li et al., 2018b*). Spinal cord tissues were rapidly removed to measure wet weight. Then at 105 °C for 48 h the tissues were dried and measure dry weight.

### TUNEL (terminal deoxynucleotidyl transferase-mediated dUTP-biotin nick end labeling) Assay

10-mm-thick sections were subjected to fluorometric TUNEL assay using the commercial kit (Beyotime, Haimen, China) to detect the apoptotic DNA strand breaks (*Bao et al., 2018*). The sections were fixed with 4% neutral-buffered formaldehyde for 20 min and incubated with proteinase K for 20 min, followed by the labeling reaction for 2 h. Then, the nuclei were stained with DAPI and images captured by fluorescence microscopy.

### Western Blotting

The protein expression levels of MAP2K3, Notch2 in segments L4–L6 of spinal cord tissues were detected with Western Blotting. Spinal cord tissues were collected at 48 h after SCII. Total proteins were extracted by using protein lysis buffer. Rabbit monoclonal anti-MAP2K3 (1:5,000, abcam), rabbit monoclonal anti-Notch2 (1:500, Santcruz), rabbit monoclonal anti-cleaved caspase-3 (1:500, abcam) and HRP-conjugated secondary antibodies (Beyotime, Haimen, China) were used.

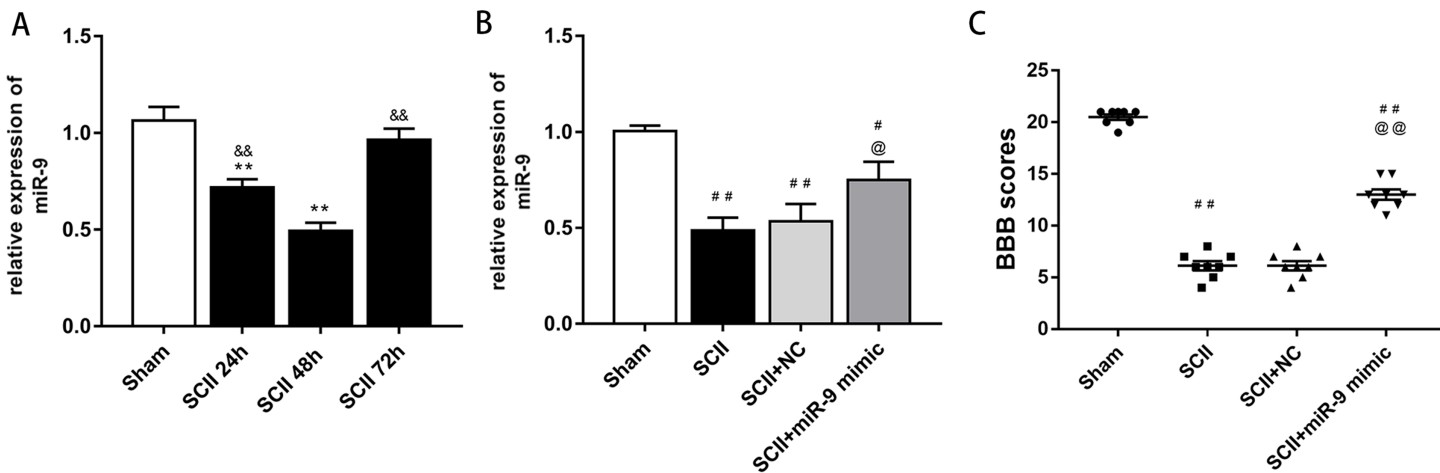

**Figure 1  miRNA expression of miR-9 after SCII.** (A) miR-9 relative expression at 24 h,48 h,and 72 h after SCII ($n = 3$). $^{*}p < 0.05$, $^{**}p < 0.01$, versus group sham, $^{\&}p < 0.05$, $^{\&\&}p < 0.01$, versus group SCII 48h. (B) miR-9 relative expression in four groups at 48 h after SCII ($n = 3$). (C) Neurological function was evaluated by means of the BBB scoring at 48 h after SCII ($n = 8$). $^{\#}p < 0.05$, $^{\#\#}p < 0.01$, versus group sham, $^{@}p < 0.05$, $^{@@}p < 0.01$, versus group SCII, for (B) and (C).           

### Enzyme-linked immunosorbent assays (ELISAs)

The rats were euthanized at 48 h after SCII. ELISA kits purchased from Signalway Antibody Company (College Park, maryland, USA) were used to test the levels of IL-6 and IL-1β according to the instructions of the manufacturer.

### Statistical analysis

Data were expressed as mean ± standard deviation and analyzed by SPSS 15.0 (IBM, USA). All variables were calculated with one-way ANOVA followed by the Newman–Keuls post hoc test. $p < 0.05$ was defined significant.

## RESULTS

### Expression of miR-9

At first, we verified the expression of miR-9 at 24, 48 and 72 h after SCII by qRT-PCR. The results showed that miR-9 levels significantly decreased at 24 and 48 h ($p < 0.01$). At 48 h after SCII, miR-9 expression demonstrated the lowest level ($p < 0.01$), as presented in Fig. 1A. These results implicated that aberrant expression of miR-9 might be related with SCII.

### miR-9 mimic improved neurological function following SCII

At 48 h after SCII, we assessed the levels of miR-9 by qRT-PCR after intrathecal injection of miR-9 mimic for 3 days. miR-9 mimic significantly increased the expression of miR-9 ($p < 0.05$) (Fig. 1B). In addition, SCII induced severe neurological deficits of lower extremities ($p < 0.01$) and miR-9 mimic improved neurological function after SCII ($p < 0.01$) according to BBB scores (Fig. 1C).

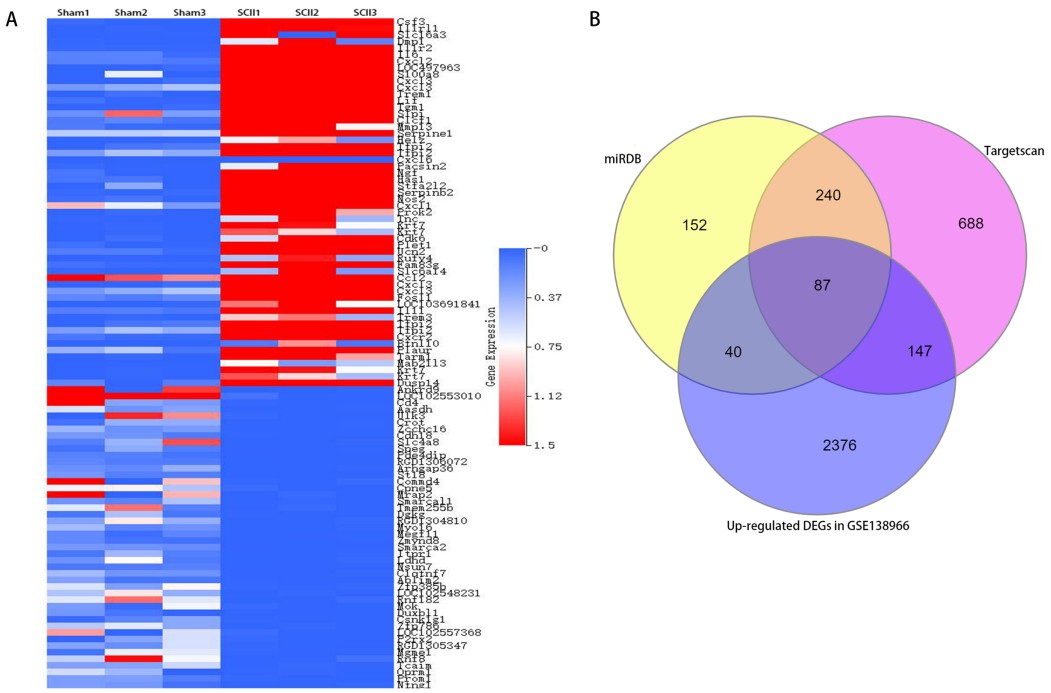

**Figure 2 A clustered heatmap of top 50 upregulated and downregulated DEGs in GSE138966 (A). Blue color signifies low expression; red color signifies high expression. Venn diagrams of potential miR-9 target genes (B).**

## miR-9 mimic attenuated BSCB leakage following SCII

Under a fluorescent microscope, EB extravasation exhibits red color (*Fang et al., 2015*; *Li et al., 2014b*). EB extravasation at 48 h after SCII was more ($p < 0.01$ versus the sham rats). In contrast, EB extravasation was reduced with miR-9 mimic pretreatment ($p < 0.01$), as depicted in Fig. 2A. EB fluorescence density and content were calculated, as depicted in Figs. 2B and 2C. Spinal cord edema was evaluated based on water content. SCII induced spinal cord edema, whereas miR-9 mimic reduced spinal cord edema ($p < 0.05$) (Fig. 2D).

## miR-9 mimic reduced apoptosis and neuroinflammation following SCII

We explored effects of miR-9 mimic on apoptosis at 48 h after SCII. Figures 3A–3B demonstrates the TUNEL results. Apoptosis rate increased following SCII ($p < 0.01$). Lower apoptosis rate was found in operated rats subjected to miR-9 mimic pretreatment compared to the SCII group ($p < 0.01$). The expression of cleaved-caspase3 protein increased at 48 h after reperfusion ($p < 0.01$), whereas miR-9 mimic significantly attenuated this downregulation ($p < 0.01$). as presented in Fig. 3C.

We assessed activation of IL-6 and IL-1β expression by ELISAs after miR-9 mimic pretreatment. The data meant that IL-6 and IL-1β were all upregulated at 48 h after SCII. Intrathecal pretreatment with miR-9 mimic evidently reduced this upregulation ($p < 0.01$). The findings are presented in Figs. 3D–3E.

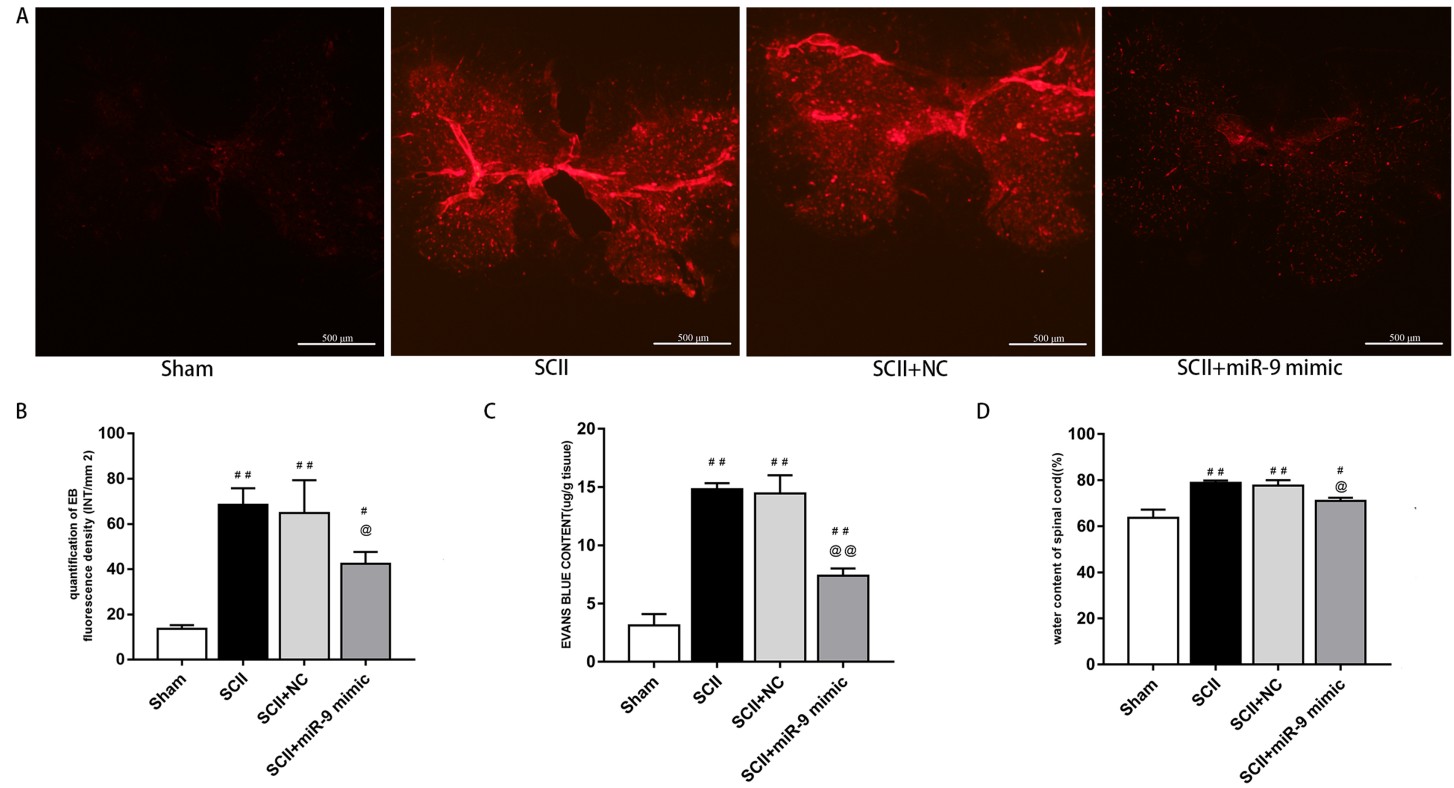

**Figure 3 Effects of on BSCB disruption and spinal cord water content at 48 h after SCII.** (A) Representative fluorescence of extravasating EB in four groups ($n = 3$). scale bar = 500 μm. (B) EB fluorescence density (INT/mm$^2$) ($n = 3$). (C) EB content (μg/g) of spinal cord ($n = 3$). (D) Percentage water content ($n = 3$). $^{\#}p < 0.05$, $^{\#\#}p < 0.01$, versus group sham, $^{@}p < 0.05$, $^{@@}p < 0.01$, versus group SCII.

## Bioinformatics analysis of potential miR-9 targets

To study the participation of miR-9, target mRNAs of miR-9 were predicted by means of databases Targetscan and miRDB. 327 potential target genes were selected for miR-9 in 2 databases (Table S1). In addition, 4,829 differentially expressed genes (DEGs) (2,650 upregulated and 2,179 downregulated) were attained with a |log2FC| ≥ 1 and $p < 0.05$ between sham-operated samples and SCII samples in GSE138966 were obtained (Table S2). A clustered heatmap of top 50 upregulated and downregulated DEGs in GSE138966 is displayed in Fig. 4A. 87 intersection genes of the up-regulated DEGs and the above predicted genes were identified as potential miR-9 target genes (Table S3 and Fig. 4B). We further conducted GO and KEGG pathway analysis to detect the key target genes of miR-9.

GO analysis was performed to investigate the enrichment of miR-9 potential target genes on biological processes. Several biological processes such as "central nervous system development", "regulation of growth" and "response to cytokine" ($p < 0.05$), were significantly enriched. The results are shown in Fig. 5A. Enrichment of miR-9 potential target genes was also conducted by KEGG pathway analysis. Several significantly representative enriched pathways for potential target genes of miR-9 were identified. The results were notably enriched in several pathways, including "Notch signaling

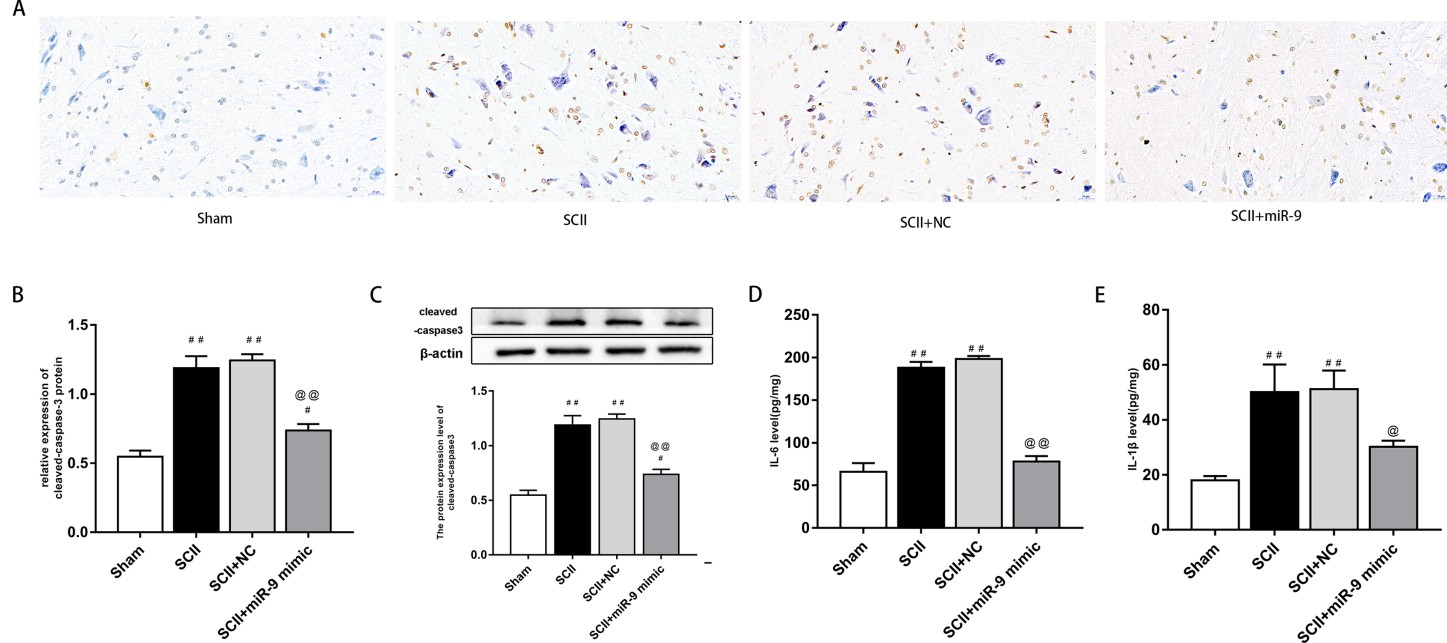

**Figure 4 miR-9 mimic reduced apoptosis rate and the production of IL-6 and IL-1β at 48 h after SCII.** (A) Representative TUNEL stain in four groups ($n = 3$). scale bar = 20 μm. (B) Apoptosis rate in four groups. (C) Effects of miR-9 mimic on the expression of cleaved caspase-3 ($n = 3$). (D–E) IL-6 and IL-1β at 48 h after SCII ($n = 3$). $^{#}p < 0.05$, $^{##}p < 0.01$, versus group sham, $^{@}p < 0.05$, $^{@@}p < 0.01$, versus group SCII.

pathway", "MAPK signaling pathway", "Focal adhesion" and "Prolactin signaling pathway" ($p < 0.05$), as shown in Fig. 5B and Table 1. The miRNA–pathway–gene network is demonstrated in Fig. 5C. It was noted that MAP2K3 and Notch2 were enriched in MAPK signaling pathway and Notch signaling pathway, respectively (Fig. S1).

## miR-9 mimic or SB203580 improves neurological function and alleviates the upregulation of cytokines after SCII

Figure 6A demonstrated the neurological function which was evaluated according to BBB scores. SCII induced neurological deficits of lower limbs. Compared to the SCII group, miR-9 inhibitor decreased BBB scores ($p < 0.01$). SB203580 enhanced neurological function recovery ($p < 0.01$). The miR-9 inhibitor + SB203580 group showed higher BBB scores ($p < 0.01$ vs SCII group). The results implied SB203580 could eliminate neurological deficits caused by miR-9 inhibitor. The protein expression of MAP2K3 increased at 48 h after SCII ($p < 0.01$), whereas miR-9 mimic notably reduced this increase ($p < 0.01$) and miR-9 inhibitor exacerbated the increase. SB203580 recuded the protein expression of MAP2K3 ($p < 0.01$) and SB203580 also reversed the effect of miR-9 inhibitor, as presented in Fig. 6B. Next, we detected the expression levels of IL-6 and IL-1β in eight groups. As shown in Figs. 6C–6D, miR-9 inhibitor increased IL-6 and IL-1β (all $p < 0.01$ vs SCII group), while SB203580 reduced the release of IL-6 and IL-1β (all $p < 0.01$ vs SCII group). In addition, IL-6 and IL-1β expression were reduced in the miR-9 inhibitor + SB203580 group compared with the miR-9 inhibitor group.

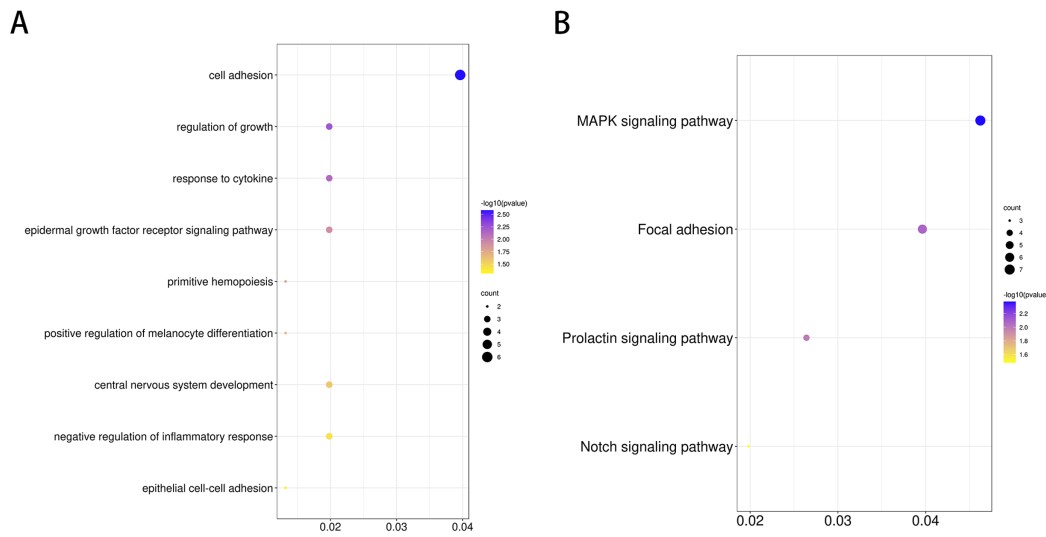

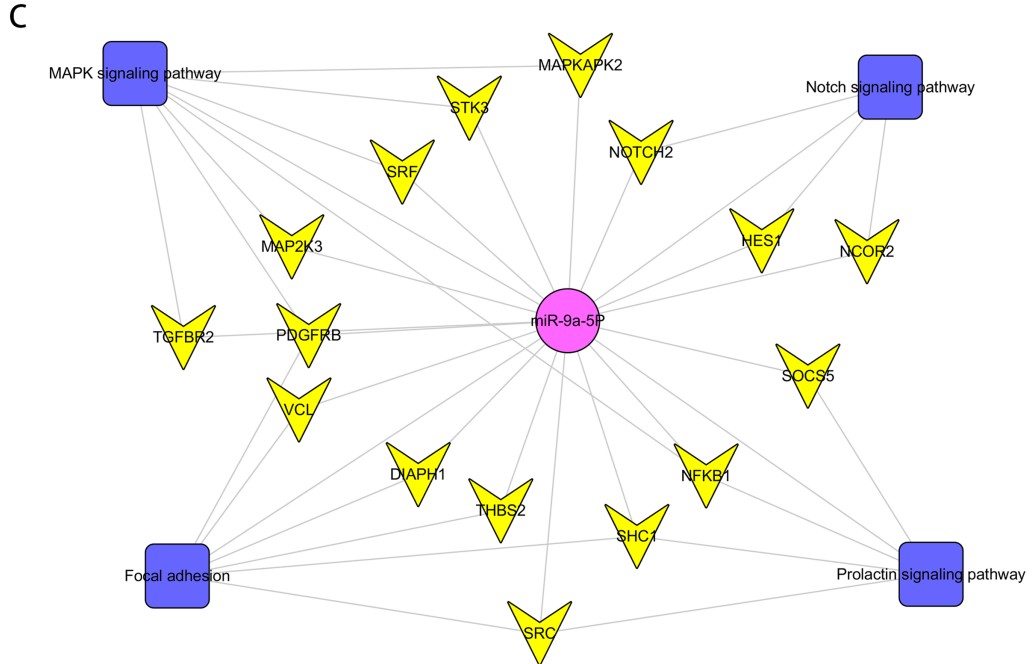

**Figure 5 Bioinformatics analysis of potential rno-miR-9 target mRNAs.** The GO annotations for biological process and KEGG pathway analysis of significant GO (A) and KEGG (B) enrichment terms. (C) The miR-pathway-gene network.

## miR-9 mimic or Nocth2 siRNA improves neurological function and reduces the protein expression of cleaved-caspase3 after SCII

Figure 7A showed the neurological function which was evaluated according to BBB scores. SCII induced severe neurological deficits of lower limbs ($p < 0.01$); meanwhile, si-Notch2 relieved neurological damage induced by SCII ($p < 0.01$), while miR-9 inhibitor aggravated neurological deterioration ($p < 0.05$). And the miR-9 inhibitor + si-Notch2

**Table 1 KEGG Pathway Terms for potential targets of miR-9 ($p < 0.05$).**

| KEGG PATHWAY Term | P value | Genes |
|---|---|---|
| MAPK signaling pathway | 0.004217403 | **MAP2K3**\*, TGFBR2, PDGFRB, NFKB1, MAPKAPK2, SRF, STK3 |
| Focal adhesion | 0.008281511 | DIAPH1, PDGFRB, SHC1, THBS2, SRC, VCL |
| Prolactin signaling pathway | 0.010146632 | NFKB1, SHC1, SOCS5, SRC |
| Notch signaling pathway | 0.033001048 | HES1, **NOTCH2**\*, NCOR2 |

**Note:**
\* MAP2K3 and Notch2 were enriched in MAPK signaling pathway and Notch signaling pathway, respectively.

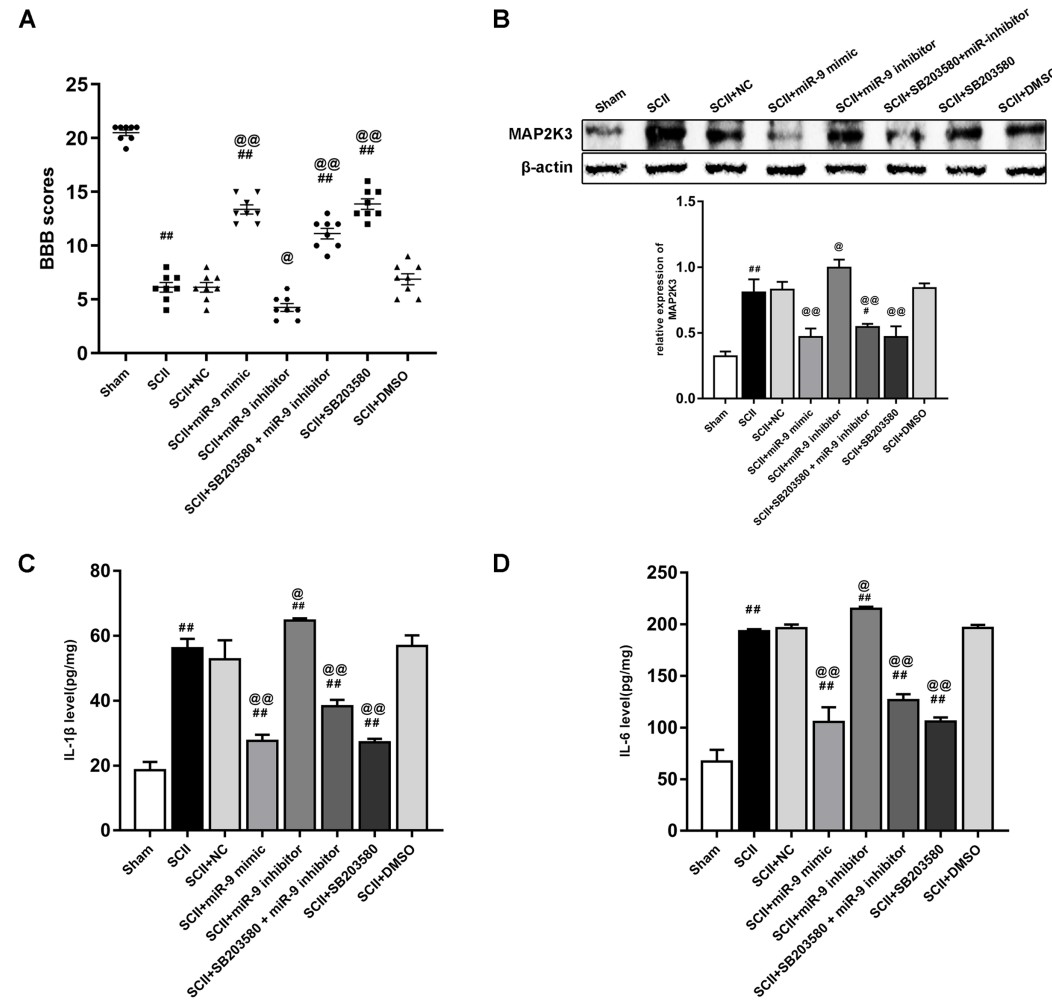

**Figure 6 Intrathecal pretreatment with miR-9 mimic or SB203580 ameliorates neurological function and alleviates the upregulation of cytokines after SCII.** (A) Neurological function scores at 48 h after SCII in eight groups. Each symbol represents one rat ($n = 8$). (B) The protein expression of MAP2K3 at 48 h after SCII in eight groups ($n = 3$). (C–D) Measurements of IL-6 and IL-1β expression levels by ELISA in eight groups ($n = 3$). $^{\#}p < 0.05$, $^{\#\#}p < 0.01$, versus group sham. $^{@}p < 0.05$, $^{@@}p < 0.01$, versus group SCII.

showed higher BBB scores ($p < 0.01$ vs SCII group) indicating si-Notch2 could eliminate the damage caused by miR-9 inhibitor. The effect of miR-9 on Notch2 protein expression was similar to that on MAP2K3 protein expression, as present in Figs. 7B–7C. Then we

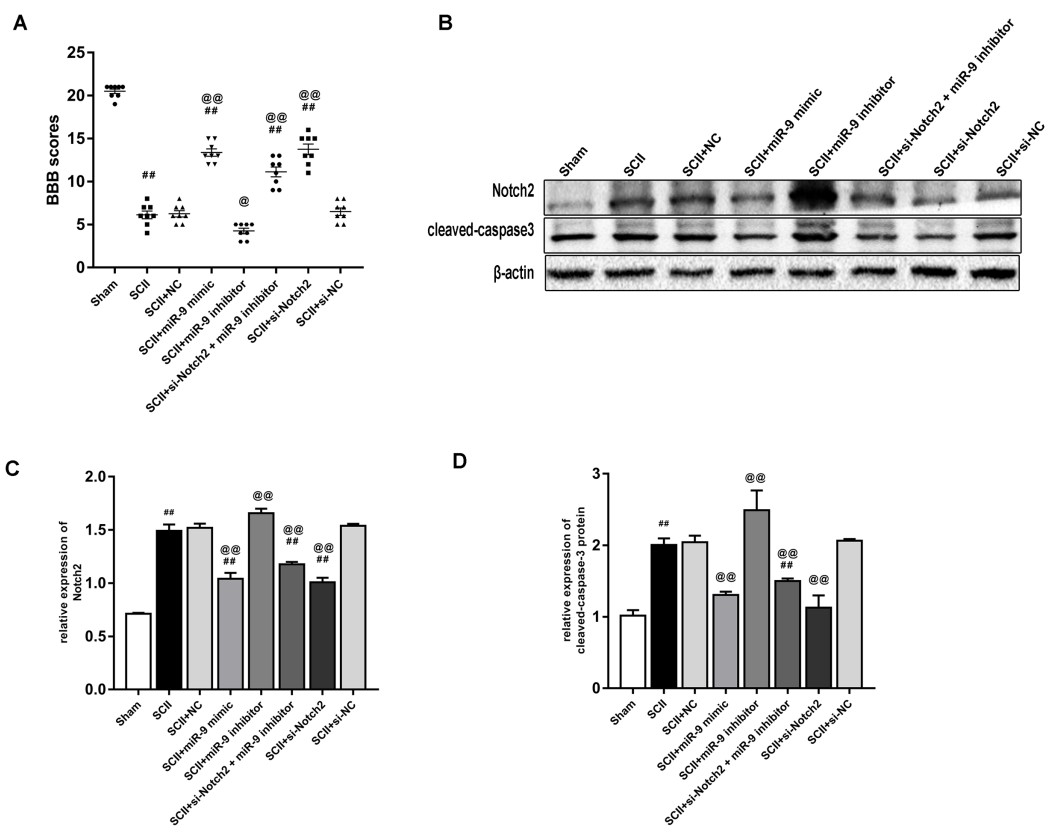

**Figure 7 Intrathecal pretreatment with miR-9 mimic or Notch2 siRNA ameliorates neurological function and reduces the increase of cleaved-caspase3 after SCII.** (A) Neurological function scores at 48 h after SCII in eight groups. Each symbol represents one rat ($n = 8$). (B–D) Representative Western blotting and the protein expression of Notch2 and cleaved-caspase3 at 48 h after SCII in eight groups ($n = 3$). $^{\#}p < 0.05$, $^{\#\#}p < 0.01$, versus group sham. $^{@}p < 0.05$, $^{@@}p < 0.01$, versus group SCII.

assessed the expression of cleaved-caspase3 in SCII. As shown in Figs. 7B–7D, intrathecal injection of Notch2 siRNA reduced the increase of cleaved-caspase3 (vs SCII group; $p < 0.01$). In contrast, miR-9 inhibitor increased cleaved-caspase3 (vs SCII group; $p < 0.01$), while cleaved-caspase3 expression were reduced in the miR-9 inhibitor + Notch2 siRNA group, and were comparable in the SCII and miR-9 NC groups.

## DISCUSSION

The present study indicated that miR-9 mimic preserved neurological function after SCII. Candidate target genes of miR-9 were collected based on bioinformatics analysis. We further found that MAP2K3 and Notch2 protein expression were reduced by miR-9 mimic. Moreover, we demonstrated that miR-9 mimic attenuated BSCB leakage, reduced apoptosis and reduced the expression of IL-6 and IL-1β after SCII. Therefore, these results suggested that miR-9 mimic reduced neurological function injury after SCII and inhibited apoptosis and neuroinflammation possibly through MAP2K3 or Notch2-mediated signaling pathway in SCII.

The etiology of SCII is multifactorial and it is initiated through induction of primary and secondary injury. SCII, which might bring the risk of paralysis and paraplegia, has been reported in the literature but continues to be a rare presentation. Several SCII case reports have been recognized and reported (*Mathkour et al., 2020*; *Nagano et al., 2009*; *Wiginton et al., 2019*). *Wiginton et al. (2019)* reported a case of posterior cervical decompression leading to complete albeit transient quadriplegia caused by SCII (*Wiginton et al., 2019*). SCII related paraplegia caused by acute type B aortic dissection also be reported. Many pathological processes including apoptosis, microglia activation, BSCB disruption, mitophagy, oxidative stress, inflammatory reactions and autophagy play important roles in the evolution of SCII (*Ha Sen Ta et al., 2019*; *Li et al., 2018a*; *Li et al., 2017*; *Liu et al., 2017*; *Zhou et al., 2018*). miRNAs are one important kind of noncoding RNAs and have emerged as novel targets for mediating numerous neurological diseases and regulating several physiological functions (*Li et al., 2019*). Studies have shown that miRNAs are differentially expressed in injured spinal cord tissues after SCII and implicate their important regulatory roles in SCII (*Hu, Lv & Yin, 2013*; *Li et al., 2016a*). We have found miR-186-5p mimic significantly reduced neuroinflammation following SCII partly though reducing the induction of CXCL13, TLR3 or wnt5a (*Chen et al., 2020a*). miR-199a-5p could alleviate SCII-induced apoptosis via targeting of ECE1 (*Bao et al., 2018*). Upregulation of miR-129-5p alleviated SCII by reducing inflammation-related BCSB and neuronal damage by inhibiting TLR3 and HMGB1-associated cytokines (*Li et al., 2017*). The molecular mechanisms of miRNAs causing SCII remains largely elusive because of their complex regulatory network in SCII.

miR-9 is highly expressed in CNS and was dysregulated in different neurodegenerative disease, such as Alzheimer's disease, Huntington's disease and stroke (*Lukiw, 2007*; *Packer et al., 2008*; *Wei et al., 2016*). miR-9 alleviated ischemic injuries by reducing anti-cardiomyocyte apoptotic affects via targeting KLF5 (*Yang et al., 2019*). Overexpression of miR-9 alleviated ischemia injury induced by middle cerebral artery occlusion and regulated the process of autophagy by targeting ATG5 expression (*Wang et al., 2018*). Upregulation of miR-9 ameliorated NLRP1 inflammasome-mediated ischemic injury in rats following ischemic stroke (*Cao et al., 2020*). In our study, we found that miR-9 levels apparently decreased after SCII and miR-9 mimic improved neurological function following SCII. However, a previous study showed that miR-9 expression was upregulated in the spinal cord of the amyotrophic lateral sclerosis (ALS) transgenic mice (*Zhou et al., 2013a*). The difference of expression trend may be due to the fact that miR-9 regulates neurogenesis by acting on neural or non-neural cell lineages with different model systems. In the present study, miR-9 might function through neurons. miR-9 might be associated with the proliferation and differentiation of neural stem cells (NSCs) and neural progenitor cells (NPCs) in ALS. In this study, GO analysis was performed to investigate the enrichment of miR-9 potential target genes on biological processes. Several biological processes such as "central nervous system development", "regulation of growth" and "response to cytokine" ($p < 0.05$), were significantly enriched. KEGG pathway analysis implied several significantly representative enriched pathways for potential target genes of miR-9 were identified, such as MAPK and Notch pathways. It was noted that MAP2K3 and Notch2 were enriched in

MAPK signaling pathway and Notch signaling pathway, respectively. MAPK signaling pathway and Notch signaling pathway have been studied in CNS. miR-21 exerted its protective effect against blood brain barrier (BBB) disruption by blocking the MAPK signaling pathway via targeted inhibition of MAP2K3 (*Yao, Wang & Zhang, 2018*). Neurotrophin-3 inhibited the content of TNF-β, IL-6 and IL-1β in spinal cord injury through inhibiting the MAPK signaling pathway (*Ye et al., 2020*). Metformin reduced microglial activation and inhibited the production of pro-inflammatory cytokines including IL-6, IL-1β and TNF-α via MAPK and NF-κB signaling pathway to improve neurobehavioral function following traumatic brain injury (*Tao et al., 2018*). Inhibition of Notch2 reduced cerebral I/R-induced cell death in the short term (*Meng et al., 2015*). miR-485-5p inhibited neuron apoptosis following I/R injury via targeting Rac1/Notch2 signaling (*Chen et al., 2020c*). In our study, we found that miR-9 mimic markedly attenuated this upregulation of MAP2K3 and Notch2. Moreover, miR-9 mimic inhibited the increase of IL-6 and IL-1β induced by SCII. Furthermore, miR-9 mimic reduced the expression of cleaved-caspase3 and apoptosis rate after SCII. In addition, miR-9 inhibitor increased IL-6 and IL-1β after SCII, while SB203580 reduced the upregulation of IL-6 and IL-1β. IL-6 and IL-1β expression were reduced in the miR-9 inhibitor+ SB203580 group compared to the miR-9 inhibitor group. Notch2 siRNA reduced the increase of cleaved-caspase3 after SCII. miR-9 inhibitor increased cleaved-caspase3, while cleaved-caspase3 expression were reduced in the miR-9 inhibitor+ Notch2 siRNA group. MAP2K3 might be the target gene of miR-9 for involving neuroinflammation after SCII. Notch2 might be the target gene of miR-9 for regulating apoptosis after SCII.

The limitation of the present study is that we only explored the expression of miR-9 during 72 h following SCII. We found that miR-9 levels significantly decreased at 24 and 48 h. At 48 h after SCII, miR-9 expression demonstrated the lowest level. The expression level of miR-9 returned to the Sham group level at 72 h after SCII. Longer time course is needed to detect for further exploring the roles of miR-9 after 72 h in SCII.

In conclusion, miR-9 mimic preserved hind limb function after SCII through reducing apoptosis and neuroinflammation. Also, miR-9 mimic might protect against SCII though MAP2K3-mediated neuroinflammation and Notch2-mediated apoptosis.

### Funding
The authors received no funding for this work.

### Competing Interests
The authors declare that they have no competing interests.

### Author Contributions
- Fengshou Chen conceived and designed the experiments, analyzed the data, prepared figures and/or tables, authored or reviewed drafts of the paper, and approved the final draft.

- Jie Han performed the experiments, authored or reviewed drafts of the paper, and approved the final draft.
- Xiaoqian Li performed the experiments, analyzed the data, prepared figures and/or tables, and approved the final draft.
- Zaili Zhang performed the experiments, authored or reviewed drafts of the paper, and approved the final draft.
- Dan Wang conceived and designed the experiments, analyzed the data, prepared figures and/or tables, and approved the final draft.

### Animal Ethics

The following information was supplied relating to ethical approvals (i.e., approving body and any reference numbers):

The Ethics Committee of China Medical University provided full approval for this research (CMU 2020391).

### Data Availability

The raw measurements are available in the Supplemental Files.

### Supplemental Information

Supplemental information for this article can be found online at http://dx.doi.org/10.7717/peerj.11440#supplemental-information.

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
