# Peer review of "Identification of the biological function of miR-9 in spinal cord ischemia-reperfusion injury in rats"

_PeerJ, doi:10.7717/peerj.11440_

## Round 0.1 · original submission · Minor Revisions

Please address the concerns of the reviewers and amend your manuscript accordingly.

Reviewer 1 ·

Basic reporting

This work reports on the bioinformatics analysis and experimental validation in identifying the biological importance of miRNA miR-9 for the treatment of neurological disease Spinal cord ischemia-reperfusion injury, multiple cases of which have been already reported in humans. The study was carried out in the Rat model. The study revealed the association of miR-9 mimic with multiple signaling pathways of Spinal cord ischemia-reperfusion injury and can be a potential therapeutic agent for it. Gene Ontology and KEGG bioinformatics analyses were also performed to identify potential target genes of miR-9 that are associated with important biological processes and signaling pathways. Considering the recent developments in miRNA-based therapeutics, this present study can be considered as an important contribution to the field. This work is within the scope of the journal. The authors used the English language professionally throughout the manuscript and the manuscript was appropriately structured following PeerJ standard. Figures are of publication quality and appropriately annotated. Raw data is available as supporting information files.

To aid the readers, a little more background on the Spinal cord ischemia-reperfusion injury disease should be added. Please discuss a couple of case studies on Spinal cord ischemia-reperfusion injury (For instance: https://dx.doi.org/10.7759%2Fcureus.5279).

Experimental design

The objective of the study is well defined and experiments are well structured and seem to be performed accurately following standard strategies.

A little more background details on GO and KEGG analysis approaches should be provided in the “Materials and Methods” section, and the reasons to choose these approaches (compared to other similar approaches) should be briefly described.

Validity of the findings

The results are presented stepwise in a logical manner and also in detail. Statistical analyses have been performed and reported. The results are well discussed answering the original research question.

A separate “Conclusions” section should be added to summarize the results and any future application of this workflow and applicability of this work should be discussed.

·

Basic reporting

Basic reporting
1. I thank the authors for their efforts to write the manuscript in clear and professional English. However, the manuscript could use grammar and language check. Grammatical errors are seen in multiple places. Some examples are lines 34-37, 211-213, 242-244, 260-261.
2. I appreciate the authors’ efforts to set the context for the study in the introduction section. The manuscript could benefit from further elaboration about why the authors chose to study mir-9 compared to other microRNAs like mir21 that might be involved in SCII and other neurodegenerative diseases. Much of this context and background is provided in the discussion section (lines 229-240). A better place for this background would be the introduction.
3. Although the authors reference multiple papers related to mir9’s role in neuronal injury and disease, they leave out a previous paper on mir9 that investigates its role in ALS- Zhou et al, 2013 have shown that mir9 has a important role in the pathogenesis of an ALS model. Interestingly that paper showed that mir9 is upregulated in the ALS model as opposed to the downregulation seen in SCII that the authors report in the current manuscript. The authors could discuss and shed some light (even if just speculation) as to what these differences could mean in terms of the biological function of mir9 in these situations.
4. Font sizes within figures need to be larger for clarity. For examples in figure 3A, the font is very small and difficult to read. Figure and table legends need to be more descriptive. Example : Table 1 has no legend description other than a short title for each section.
5. I commend the authors’ efforts to summarize MAPK and Notch2 signaling pathways in figure 4. However, it is not original data or model of a mechanism being uncovered in this paper. It appears to be a summary of current knowledge of MAPK and Notch signaling pathways. This would be unnecessary and can simply be referenced.

Experimental design

6. The authors report that mir9 expression is downregulated 48h hours after SCII, but don’t discuss the 72 hour timepoint at which mir9 expression almost goes back up to normal compared to sham group. Is this time point significantly different from sham? It would be beneficial to know what happens after the 72 hours time point. Does the expression of mir9 go up and stay high after 72 hr? If that is the case, the downregulation of mir9 seen after SCII is pretty short lived. What would this mean for the pathology of SCII and the benefit of therapeutic intervention by targeting mir9?
7. Major comment - While this manuscript uncovers a possible mechanism of action of mir9 in spinal cord ischemia-reperfusion injury (SCII), it does not investigate this mechanism in a rigorous and compelling manner. The authors find the possibility that mir9 might be acting through a MAPK (MAP2K3) pathway or Notch pathway to confer reduced BSCB disruption, inhibit apoptosis and show reduced inflammatory response after SCII, but do not clearly show that mir9 acts through either of these pathways to exert these effects.
A compelling experiment that would test this possibility is to determine if a mir9 inhibitor shows the opposite effect to mir9 mimic on neuroinflammation and apoptosis, and these effects are rescued by a MAPK inhibitor. For examples, Yao et al showed that mir21 has a neuroprotective role after SCII via a MAPK signaling pathway by targeted inhibition of MAP2K3. In this paper they did a series of experiments utilizing mir21 mimics, inhibitors and MAP2K3 inhibitor SB203580. The authors of the current paper could utilize this or a similar MAPK inhibitor to definitively test if mir9 acts through MAP2K3. Additionally, Yao et al showed that mir21 interacts with the 3’UTR of MAP2K3 using a luciferase reporter assay. The current manuscript could use an additional experiment to validate the authors’ hypothesis that mir9 acts by downregulation of MAP2K3.
Similarly, they could show the involvement of Notch2 in this pathway by inhibition of Notch2 using siRNA as done by Meng et al, 2015.
8. Acronymns/abbreviations are not expanded the first time they are used. For example BSCB, KEGG. GSE138966 – What is it? This is not explained clearly to a non-expert reader the very first time it is mentioned in the manuscript in lines 102-104
9. Lines 171-73 could use further explanation about the rationale for conducting bioinformatic analysis.
10. Supplemental table 1 gives a list of 327 potential target genes of mir9 apparently from two different databases. It would be helpful to include additional information in this table other than just the gene name such as – the database that it came from, the selection criteria used by the database, scoring if any, description of the gene name, etc

Validity of the findings

11. The data provided are robust and clearly presented. I commend the authors for their work and effort to present the complete raw and analyzed data.
12. A separate conclusion section is not provided. There is a short summary at the end of the discussion section. The authors do not overstate the findings. However, the authors draw the conclusion that mir9 may be a potential target for reducing neuroinflammation and apoptosis after SCII. The results certainly present a possibility for it but it remains to be determined whether mir9 really acts through the proposed pathways. It could be a little premature to state that mir9 could be a potential target for therapy before a mechanism of action is established.

Additional comments

This manuscript presents an interesting possibility for a mechanism by which mir9 affects neuronal outcomes after SCII. While the data in this manuscript suggests a mechanism of action for further investigation, it does not establish this mechanism in a rigorous and compelling manner. The authors could have done a few more experiments to take it to the finish line. However, the data presented are sound and the authors have done their part in presenting all of the data. On its own, this manuscript is acceptable with revisions as suggested below.

---

## Round 0.2 · accepted · Accept

All issues pointed by the reviewers were addressed and the manuscript was revised accordingly. Therefore the amended version is acceptable now.